# Experimental Infection and Genetic Characterization of Two Different Capripox Virus Isolates in Small Ruminants

**DOI:** 10.3390/v12101098

**Published:** 2020-09-28

**Authors:** Janika Wolff, Jacqueline King, Tom Moritz, Anne Pohlmann, Donata Hoffmann, Martin Beer, Bernd Hoffmann

**Affiliations:** Institute of Diagnostic Virology, Friedrich-Loeffler-Institut, Federal Research Institute for Animal Health, Südufer 10, D-17493 Greifswald-Insel Riems, Germany; janika.wolff@fli.de (J.W.); jacqueline.king@fli.de (J.K.); Tom.Moritz@gmx.de (T.M.); anne.pohlmann@fli.de (A.P.); donata.hoffmann@fli.de (D.H.); martin.beer@fli.de (M.B.)

**Keywords:** capripox, sheeppox, goatpox, SPPV, GTPV, field strain, vaccine strain, nanopore sequencing, next-generation sequencing, MinION

## Abstract

Capripox viruses, with their members “lumpy skin disease virus (LSDV)”, “goatpox virus (GTPV)” and “sheeppox virus (SPPV)”, are described as the most serious pox diseases of production animals. A GTPV isolate and a SPPV isolate were sequenced in a combined approach using nanopore MinION sequencing to obtain long reads and Illumina high throughput sequencing for short precise reads to gain full-length high-quality genome sequences. Concomitantly, sheep and goats were inoculated with SPPV and GTPV strains, respectively. During the animal trial, varying infection routes were compared: a combined intravenous and subcutaneous infection, an only intranasal infection, and the contact infection between naïve and inoculated animals. Sheep inoculated with SPPV showed no clinical signs, only a very small number of genome-positive samples and a low-level antibody reaction. In contrast, all GTPV inoculated or in-contact goats developed severe clinical signs with high viral genome loads observed in all tested matrices. Furthermore, seroconversion was detected in nearly all goats and no differences concerning the severity of the disease depending on the inoculation route were observed. Conclusively, the employed SPPV strain has the properties of an attenuated vaccine strain, consistent with the genetic data, whereas the GTPV strain represents a highly virulent field strain.

## 1. Introduction

The genus *Capripoxvirus* within the Poxviridae family consists of three species: sheeppox virus (SPPV), goatpox virus (GTPV) and lumpy skin disease virus (LSDV) [1]. Infections of ruminants by capripox viruses induce diseases of great ecological impact. Guidelines by the World Organization for Animal Health build a framework for regulatory measures [2]. Morbidity as well as mortality are very variable depending on the immunological status of the host. Whereas morbidities of less than 10% are observed in endemic regions with a stable enzootic situation [3], morbidities up to 70–100% in susceptible populations have been described [4,5]. Mortality rates of SPPV and GTPV range from 5 to 10% in local breeds of endemic regions [6] to 100% in naïve populations [4,6,7]. Several factors can influence severity of the disease, such as breed and age of the susceptible host, immunological status, and stage of production of the host as well as the virus isolate [8,9,10].

SPPV and GTPV are highly contagious diseases [4,10,11] transmitted via aerosol [3,12,13], direct contact of infected and non-infected animals [12], and contact with contaminated feed, wool, and other objects [4]. In contrast to the mainly mechanical transmission of LSDV via insect vectors [14,15,16,17], insects do not seem to play a major role in the transmission of SPPV and GTPV [13].

Disease after infection with SPPV or GTPV is characterized by the formation of papules and nodules in the skin [3,8,10,11]. Pox lesions may be restricted to a few papules, often observed in enzootic regions [11], or can generalize and affect up to 50% of the skin surface [3,11,18]. Additionally, development of characteristic pox lesions in the respiratory and gastrointestinal tract [11,18] as well as on mucous membranes [18] is described. Furthermore, diseased animals can develop fever [3,11,18] and systemic symptoms including coughing, diarrhea, depression, emaciation, abortion [18], as well as temporary or permanent infertility [9], rhinitis, conjunctivitis, and excessive salivation [11].

Most SPPV strains cause severe clinical disease in sheep and only sub-clinical to mild clinical courses are observed in goats. GTPV in goats likewise induce substantial disease, while sheep develop, if any, mild clinical signs [4,11,13,18]. However, some isolates are able to infect both species and cause severe symptoms in sheep as well as goats [8,9,19,20]. Moreover, European breeds are more susceptible to capripox virus infections compared to African and Asian sheep, goats, and cattle [4,9,13].

Clinically, SPPV and GTPV cannot be distinguished [18]. Since there is only a single serotype of capripox viruses, serological [11,18,21] as well as antigenic [11,21] distinction between SPPV, GTPV, and LSDV is not possible. The current gold standard for indirect diagnostics is the virus neutralization assay [11]. Additionally, PCR based assays display reliable, sensitive, and rapid molecular diagnostic tools for capripox virus detection [11]. For differentiation between SPPV and GTPV, a few assays have been published. Lamien et al. developed a PCR system for distinction of SPPV and GTPV based on a 21-nucleotide deletion in the SPPV genome compared to GTPV [22]. In addition, a gel-based duplex PCR assay [23], a real-time PCR assay containing a snapback primer and a dsDNA intercalating dye [24] as well as a real-time high-resolution melting PCR assay [25] are already published.

For the profound genetic understanding and comparison of both SPPV and GTPV, full-genome sequencing is indispensable. Although individual genes have been previously characterized with Sanger sequencing methods, the availability of full genomes of the respective large 150 kb dsDNA [26] is limited, highlighting the importance of additional sequencing work to create a base line for vaccine development and diagnostic testing. The MinION platform (Oxford Nanopore Technologies, ONT) allows long reads and real-time sequencing, while the Illumina platform provides shorter, higher quality sequences in greater abundance. This combination has previously been successfully utilized for sequencing of large dsDNA viruses, e.g., African swine fever virus [27]. By combining both platforms, high quality full genome sequences can be achieved.

Here, we examined the clinical signs as well as molecular and serological response of sheep to a SPPV strain (SPPV-“V/104”) and goats to a GTPV (GTPV-“V/103”) strain, respectively, in order to characterize these strains in vivo. Possible adverse effects after inoculation of a vaccine strain as well as establishment of a challenge model were studied. During the animal trial, body temperature and clinical scores were documented and different matrices (EDTA-blood, serum, nasal swabs, and oral swabs) were taken for molecular and serological analyses. Thereby, molecular examination methods were validated internally, and different serological tests were compared regarding sensitivity and specificity. Furthermore, different inoculation routes were analyzed concerning viremia, virus shedding, and serological reaction of animals. While the goats infected with GTPV-“V/103“showed severe clinical signs independently of the inoculation route, SPPV-“V/104“ did not lead to clinical signs in the inoculated sheep. In addition, both respective strains underwent high-throughput full-genome sequencing on both a MinION and Illumina platform to receive full genomes of high quality allowing the correlation of phenotype data from the animal trials with the determined genotype.

## 2. Materials and Methods

### 2.1. Animals

Eight female sheep (cross-breed of common German breeds) between six and seven months of age and eight male goats (Weisse Deutsche Edelziege) between six and eight months old were housed in the facilities of the Friedrich-Loeffler-Institut, Insel Riems, Germany, under biosafety level 3+ conditions. The respective experimental protocols were reviewed by the state ethics commission and approved by the competent authority (State Office for Agriculture, Food Safety, and Fisheries of Mecklenburg-Vorpommern, Rostock, Germany; Ref. No. LALLF M-V/TSD/7221.3-2.1-022/10, approval date: 30 August 2010).

### 2.2. Experimental Infection and Collection of Samples

The sheep and goats were inoculated with a SPPV strain (SPPV-“V/104”) and a GTPV strain (GTPV-“V/103”), respectively. Both strains were part of the historic collection of the German National Reference Laboratory for Sheep Pox and Goat Pox. Documentation about the origin of the strains was not available anymore. Therefore, in order to analyze the relations to other known capripox virus strains, both isolates were sequenced to achieve full-length genomes (for detailed information see Sequencing of SPPV-“V/104” strain and GTPV-“V/103” strain). For each virus isolate, three animals of the respective target species were inoculated with 3 mL of the virus intravenously (i.v.) and 1 mL subcutaneously (s.c.), another three animals received the virus intranasally (i.n.), and the final two animals were kept as in-contact controls (i-c). Virus propagation was performed on SFT-R cells (FLI cell culture collection number CCLV-RIE0043). Therefore, SFT-R cells with confluence of approximately 90% were infected with either GTPV-“V/103” or SPPV-“V/104” and incubated for 7 days at 37 °C. Afterwards, infected cells were frozen at −80 °C, and virus-cell-suspension was stored at −80 °C until further usage. Virus re-titration on SFT-R cells revealed a titer of 10^5.75^ tissue culture infectious dose_50_ (TCID_50_)/mL for the SPPV-“V/104” strain and a titer of 10^6.13^ TCID_50_/_mL_ for the GTPV-“V/103” strain. During the animal trial, EDTA-blood for the evaluation of cell-associated viremia, serum for serological analyses and examination of cell-free viremia, as well as oral and nasal swabs were taken at different time points: −1 dpi, 3 dpi, 5 dpi, 7 dpi, 10 dpi, 13 dpi, 15 dpi, 18 dpi, 21 dpi, 24 dpi, and 28 dpi. Body temperature was measured daily from −4 dpi until 28 dpi, clinical reaction scores were documented daily from 1 dpi until 28 dpi. Therefore, a modified clinical reaction score system [28] of Carn and Kitching [29] was used for evaluation. During necropsy, several organ samples (mandibular lymph node (ln), cervical ln, mediastinal ln, mesenterial ln, tonsil, spleen, liver, lung, heart, kidney, salivary gland, testicles, oral mucosa, thigh muscle, and other conspicuous organs as well as skin nodules of different locations) of experimentally infected and in-contact goats were taken and analyzed regarding the presence of viral genomes. Since no capripox virus genome was detected in the inoculated and the in-contact sheep from 7 dpi on, we did not take organ samples of sheep during autopsy.

### 2.3. DNA Extraction of Different Samples and Pan Capripox Real-Time qPCR Analysis

Before DNA extraction, organ samples were homogenized in serum-free medium using the TissueLyser II (Qiagen, Hilden, Germany). DNA extraction of homogenized organ samples, EDTA-blood, serum, oral, and nasal swabs was performed using the NucleoMag Vet kit (Macherey-Nagel, Düren, Germany) according to the manufacturer’s instructions with the King Fisher Flex (ThermoScientific, Darmstadt, Germany), with the following modifications of volume for non-organ samples: 10 µL Proteinase K, 10 µL NucleoMag B-Beads, 300 µL washing buffer VEW 1, 300 µL washing buffer VEW 2, and 300 µL 80% ethanol. To control successful DNA extraction, an internal control DNA was added to the samples [30]. The pan capripox real-time qPCR was performed as described by Bowden et al. [18], but extended with a modified Capri-p32 TaqMan probe [31] using the PerfeCTa ToughMix from Quanta BioSciences (Gaithersburg, MD, USA).

### 2.4. Serological Analyses

For serological analyses, the serum neutralization test (SNT) and a commercially available ID Screen Capripox double antigen (DA) ELISA (ID.vet, Montepellier, France) were used.

For the SNT, heat-inactivated (56 °C, 30 min) serum samples were diluted in serum-free medium in log2 dilution series from 1:10 to 1:1280 in a 96 well plate. Each dilution step was analyzed in triplicates. Subsequently, 50 µL of a LSDV-Neethling vaccine strain with a titer of 10^3.3^ TCID_50_/mL were added to the samples and incubated for 2 h at 37 °C and 5% CO_2_. Afterwards, MDBK cells (FLI cell culture collection number CCLV-RIE0261) with a concentration of approx. 30,000 cells/100 µL were added. Incubation took place for 7 days at 37 °C and 5% CO_2_. Examination of the antibody titer was performed under a light microscope (Nikon Eclipse TS-100) using the Spearman–Kärber method [32,33].

The ID Screen Capripox DA ELISA was performed following the manufacturer’s instructions. An S/P% ratio of ≥ 30 was defined as positive.

### 2.5. Sequencing of SPPV-“V/104” Strain and GTPV-“V/103” Strain

After DNA extraction of the respective strains propagated on SFT-R cells utilizing the MasterPure Complete DNA and RNA Purification Kit (Lucigen/Biozym Scientific GmbH, Hessisch Oldendorf, Germany) according to the manufacturer’s instructions and taking precautionary measures to avoid as much shearing of the DNA as possible, for example no vortexing and minimal pipetting, the samples were prepared for sequencing on both the Illumina HiSeq 2500 (Illumina, San Diego, CA, USA) and MinION platform (ONT, Oxford, UK).

For high-throughput Illumina sequencing (Illumina), samples were sent to the ISO17025 accredited Eurofins Genomics lab (Eurofins Genomics Germany GmbH, Ebersberg, Germany) and subsequently prepared for sequencing on the Illumina HiSeq 2500 platform according to the company’s workflow. A total of 5 million paired reads per sample were produced for further analyses.

For the MinION platform, library preparation with the Rapid Barcoding Kit (SQK-RBK004, ONT) was conducted according to the manufacturer’s instructions. This entails a two-step protocol starting with the ligation of specific barcodes after cleavage of the genomic DNA by a barcoded transposase complex, followed by the attachment of sequencing adapters after sample pooling. An ONT MinION sequencing device (Mk1B, ONT) in combination with a MinIT (MT-001, ONT) for live basecalling and a R9.4.1 Flow Cell (FLO-MIN106D, ONT) were utilized for real-time sequencing. To receive the highest quality reads possible, high accuracy basecalling with the basecaller Guppy (v.3.2.9, ONT) was chosen for a 12-h run. Guppy was also used for demultiplexing and trimming of reads. After trimming, demultiplexing and quality check of the produced reads, consensus sequence generation was conducted in an iterative mapping approach with Geneious v.11.1.5 (Biomatters, New Zealand).

The respective full genome sequences obtained in this study were submitted to the INSDC under accession MW020570 (GTPV-“V/103”) and MW020571 (SPPV-“V/104”).

For phylogenetic analyses, sequences were aligned using MAFFT [34]. Subsequently, maximum likelihood analyses using RAxML [35], including 1000 bootstrap replicates were performed.

## 3. Results

### 3.1. Clinical Signs after Experimental Infection

No marked changes in body temperature could be observed for sheep inoculated with the SPPV-“V/104” strain or the in-contact control animals. Additionally, SPPV-“V/104” did not lead to clinical signs in any sheep.

In contrast to the sheep, all goats inoculated with the GTPV-“V/103” strain developed high fever up to 41.6 °C starting at 1 dpi and lasting for several days, irrelevant of the used inoculation method. Interestingly, both i-c control animals also developed a fever, however starting at 18 dpi (Z/259) and 24 dpi (Z/260), indicating the beginning of an infection with GTPV.

All goats, regardless of the inoculation method, developed clinical signs including ocular and nasal discharge and characteristic lesions of the skin (Figure 1), but with clear differences between the two inoculation methods as well as i-c animals concerning the time points of the onset of clinical signs (Figure 2).

First clinical signs were seen in the three goats inoculated i.v. + s.c. Here, the clinical reaction started at day 2 pi with mild signs including slightly reduced feed intake, mild nasal discharge, and development of sporadic papules. At 7 dpi, severe clinical signs (reduced activity and feed intake, strong coughing, and severe nasal discharge as well as numerous skin lesions) were observed, resulting in the euthanasia of all three animals at 10 dpi due to ethical reasons. One of three goats inoculated i.n. also developed severe clinical signs, including generalized skin lesions, severe coughing, and nasal discharge as well as clearly reduced activity and feed intake, and had to be euthanized at 15 dpi (Z/256). However, both remaining goats inoculated i.n. showed only a mild to moderate clinical response, starting at 9 dpi (Z/258) and 10 dpi (Z/257), respectively. Here, slightly reduced feed intake and activity, development of skin lesions and slight respiratory signs could be observed. At day 20 pi, first clinical signs (small number of papules as well as slightly reduced activity) could be observed for i-c goat Z/259, whereas i-c goat Z/260 showed first clinical signs (slightly reduced activity and feed intake, slight nasal discharge as well as development of some papules) 24 days after the start of the inoculation (Figure 2). During the following days, both in-contact animals became severely affected and likewise had to be euthanized due to ethical reasons. Transmission of GTPV via direct contact or aerosol was therefore clearly demonstrated.

### 3.2. Replication and Virus Shedding

In the SPPV group, viral genome could be detected in only two nasal swabs and a single oral swab sample of two animals (S/652, S/673), with very high Cq values between 36.3 (S/652, i.n., 5 dpi, oral swab) and 38.0 (S/673, i.n., 3 dpi, nasal swab), and detection at 3 dpi and 5 dpi (Table 1). Since all other samples were tested negative for capripox virus genome (Table 1), no conclusion can be drawn regarding a possible correlation between viral replication, shedding and the route of infection.

First positive PCR results in the GTPV group were detected at 3 dpi in the intranasally inoculated goats with high Cq values of 37.7 (Z/257, nasal swab) and 38.1 (Z/258, nasal swab). From 5 dpi on, viral genome was detected in all inoculated goats in at least one sample type. Cq values within and between the i.v. + s.c. group and the i.n. group were very similar and ranged from 18.9 (Z/256, i.n., nasal swab, 10 dpi) to 38.1 (Z/258, i.n., nasal swab, 3 dpi). All goats euthanized due to ethical reasons were tested positive for capripox virus genome in all four matrices at the day of euthanasia. Furthermore, EDTA-blood scored positive earlier or at the same day as serum samples with consistently lower Cq-values. Nasal swabs proved to be a more sensitive sample for the detection of viral genome than oral swabs (Table 2). Intranasally inoculated animals were still positive for substantial viral DNA levels at 28 dpi in nasal swabs. Interestingly, viremia started earlier in i.v. + s.c. inoculated goats compared to goats inoculated intranasally, whereas nasal as well as oral swab samples tested positive in i.n. inoculated goats prior to goats of the i.v. + s.c. group, indicating a correlation between the site of initial viral replication and infection route (Table 2). First positive results for i-c goats were detected at 5 dpi (Z/260, oral swab, Cq 35.8), and consistent viral replication could be observed in swab samples from 10 dpi (Z/260) and 13 dpi (Z/259) on (Table 2).

### 3.3. Viral Genome Loads in Organ Samples

During necropsy of the goats infected with GTPV-“V/103”, a panel of organs was taken and analyzed using the pan Capripox real-time qPCR. In severely affected goats, most organs taken tested positive for capripox virus genomes (Table 3), clearly indicating a generalized infection with GTPV. In addition to organs of the respiratory tract, for example lung tissue (e.g., Z/253, 21.9), conchae (e.g., Z/256, 17.8), or the trachea (e.g., Z/259, 19.9), and especially skin samples of affected areas showed high viral genome loads (e.g., tail base, Z/256, 16.9 or thoracic wall, Z/258, 21.3) in all goats. However, both animals that showed a milder disease of goat pox and survived until the end of the study were negative in most tested organ samples including certain lymph nodes. Only skin samples, testicles, and oral mucosa showed positive results for capripox virus genome in severely as well as moderately infected goats (Table 3).

### 3.4. Serological Evaluation

For serological examination, serum samples were taken from all animals at different time-points and analyzed with two different methods: SNT and DA ELISA.

In the SPPV-“V/104” group, serological response towards inoculation or contact to inoculated sheep turned out to be relatively homologous (Table 4). The DA ELISA as well as the SNT data were negative for seven out of eight animals during the whole animal trial. The SNT showed positive results for S/542 from 15 dpi on and the DA ELISA scored positive starting at 21 dpi (Table 4).

For the inoculated goats (Table 5), two out of three animals inoculated i.v. + s.c. scored positive for neutralizing antibodies at 10 dpi (Z/254, Z/255). Z/255 was also positive in the DA ELISA, whereas Z/254 stayed negative in this test. The severely affected goat of the i.n. inoculation group (Z/256) did not show any serological response until euthanasia at 15 dpi. For both other goats of this group (Z/257, Z/258), the SNT showed positive results starting at day 15 pi. Additionally, seroconversion of both goats could be detected using the DA ELISA, but at later time points (Z/257 at 21 dpi, Z/258 at 28 dpi). Both in-contact animals became infected later than the inoculated goats, explaining why seroconversion started at a later time point compared to the inoculated goats. Z/259 showed a positive result in the SNT at 23 dpi, but not in the DA ELISA, whereas Z/260 reacted negative in the DA ELISA as well as the SNT (Table 5).

### 3.5. Full Genome Sequencing of Strains SPPV-“V/104” and GTPV-“V/103”

The MinION platform produced 272,643 reads for the GTPV sample and 61,978 reads for the SPPV sample. First mapping analyses revealed a ratio of 13.54% (39,912 reads) GTPV reads and 9.87% (6115 reads) SPPV reads. Overall, the GTPV data averaged at a value of 11.3 mean quality and 499.8 mean read length, with the longest mapped read of 9934 nucleotides. In comparison, the SPPV data achieved a mean quality value of 10.3 and a mean read length of 321.8, with the longest mapped read of 5506 nucleotides.

All MinION and Illumina data were used in a hybrid assembly iterative mapping approach to gain full length, high quality consensus sequences. For SPPV-“V/104”, a total of 842,075 reads of 14,084,051 were mapped (5.98%) yielding a mean coverage of 866.4-fold. For GTPV-“V/103” a total of 613,264 reads of 14,068,961 total reads were mapped (4.36%) with a mean coverage of 701.6-fold. Consensus sequences were compared to SPPV and GTPV from the INSDC databases, respectively. SPPV-“V/104” was found 99.9% identical to accession AY077834 SPPV strain NISKHI, an attenuated vaccine strain established in the Scientific Research Agricultural Institute (Gvardeiskiy, Kazakhstan) in 2000 [26] (Figure 3A), which is in line with the absence of clinical signs in the animal trial. In addition, the ankyrin repeat protein genes 145 and 148 of the SPPV-“V/104” genome are truncated. Similar truncated genes 145 and 148 could be also identified in the SPPV-NISKHI vaccine strain [36]. This finding, together with the results of the in vivo study, support the classification of this strain in the group of the attenuated SPPV vaccines. On the other hand, GTPV-“V/103” was 100% identical to GTPV strain “India” (accession MN072620 [36]) (Figure 3B), which genome sequence could therefore be confirmed by our findings.

## 4. Discussion

### 4.1. Pathogenesis in Sheep after Inoculation with the SPPV-“V/104” Strain

Surprisingly, all eight sheep, independent of the inoculation method, showed neither any changes in body temperature, nor any clinical signs after inoculation with SPPV-“V/104”, confirming genetic findings that SPPV-“V/104” might be a vaccine strain of SPPV. However, a slight increase in body temperature as well as local reactions at the inoculation site are reported after vaccination of sheep with SPPV live-attenuated vaccines [37,38], which is in contrast to our clinical results. Furthermore, viral genome could only be detected in two nasal and a single oral swab in the present animal study (Table 1). These molecular data are in contrast to the results obtained after inoculation of cattle with different LSDV live attenuated vaccines. Here, cell-free as well as cell-associated viremia could be observed sporadically, but with low amount of vaccine virus DNA. However, samples of some individual animals remained negative during the whole study [39,40,41]. The latter correlates with the results of almost all samples obtained from sheep of the present study. Nevertheless, nasal and oral swabs of the SPPV-“V/104“-inoculated sheep were positive at 3 dpi and 5 dpi in only two and one sheep, respectively. It is notable that these two sheep were inoculated intranasally. Thus, this finding could be explained by the presence of inoculation material or slight viral replication at the inoculation site.

Specific serological response could only be detected in one sheep, starting at 15 dpi (SNT) and 21 dpi (DA ELISA), respectively. All other sheep reacted negatively in the SNT as well as the DA ELISA. (Table 4). In contrast, after vaccination with a live-attenuated SPPV Romania strain, serological response in sheep is described to start around 14 days post vaccination [37,38]. Considering the clinical and molecular data, we conclude that the investigated SPPV-“V/104” strain is potentially a SPPV vaccine strain. Two explanations for the low antibody titer are possible. First, the SPPV strain used in this study is over-attenuated, no longer leading to clinical signs in the inoculated animals or to reliable serological response. This assumption is supported by the missing detection of viremia or viral shedding during the whole study. Secondly, these animals possibly developed antibody levels lower than the detection limit, but would withstand a challenge infection with a pathogenic virus strain [9].

### 4.2. Pathogenesis in Goats after Inoculation with the GTPV-“V/103” Strain

All goats of all inoculation methods became infected with GTPV, but differences in the incubation time were observed (Figure 2). At 6 dpi, all three i.v. + s.c. inoculated goats (Z253, Z/254, Z/255) already showed clinical signs characteristic for capripox virus infections. Intranasally inoculated goats developed first clinical symptoms characteristic for GTPV infections at 8 dpi. Both i-c goats also showed a severe clinical course of GTPV infection, starting at 20 dpi (Z/259) and 24 dpi (Z/260). Thus, infection most likely took place via direct contact or aerosol from the inoculated and affected goats. Clinical course and development of clinical sign is similar compared to other studies dealing with experimental infection of goats with capripox viruses (e.g., Babiuk et al., 2009; Bowden et al., 2008). In these two studies, goats were inoculated intradermally and first clinical signs were seen at 2 and 4 dpi, respectively, by formation of a skin nodule at the site of inoculation [8,18], followed by a fever and secondary skin lesions around 6 dpi [18]. In the presented animal experiment, start of clinical symptoms was dependent on the inoculation route. In detail, first clinical signs could be observed within the first 2–6 days pi in the i.v. + s.c. group consistent with already published results. In addition to skin nodules detected in all goats with different manifestations (from partially on sparsely haired regions to generalized forms), the animals emaciated, and ocular and nasal discharge was observed, as seen in the animal trials of Bowden et al. and Babiuk et al. with different capripox virus isolates [8,18]. Six out of eight animals, independent of the inoculation route, were severely affected by goat pox and had to be euthanized during our study.

Capripox virus genome was detected at 3 dpi in nasal swab samples of two i.n. inoculated goats (Z/257, Z/258). Viremia could be observed starting at 5 dpi in all three i.v. + s.c. incolated animals and from 7 dpi and 10 dpi on in goats inoculated i.n (Table 2). These findings correlate well to those of Bowden et al., who saw viral genome load in the blood of experimentally infected goats starting from 4 dpi [18]. At 5 dpi, all experimentally infected animals of our study scored positive in the pan Capripox real-time qPCR in at least one of the four tested matrices, independent of the inoculation route. However, i.v. + s.c. inoculated animals showed viremia some days before viral shedding, whereas viral genome in nasal and oral fluid could be detected earlier than viremia in all goats of the i.n. group, indicating an influence of the route of infection towards the onset of viral replication. Both i-c goats became infected with goat pox, as verified by molecular data. Here, pattern of viral genome load in the four tested matrices is similar to goats experimentally inoculated i.n., starting with viral shedding followed by viremia (Table 2). This also fits nicely to previous publications reporting that GTPV is mainly transmitted via direct contact or aerosol [3,12,13]. Similar to the goats in the study of Babiuk et al., in which viral genome load could be detected in nasal, oral, and conjunctival swabs until the end of the study at 35 dpi [8], both surviving animals of our trial stayed positive in oral and nasal swabs until the end of the study at 28 dpi. Moreover, nasal swabs turned out to be more sensitive in our study than oral swabs (Table 2). After necropsy, viral genome load of different organs was also tested using the pan Capripox real-time qPCR. As described before [18], highest viral genome loads could be detected in the skin samples taken from affected areas. Furthermore, organs of the respiratory and gastrointestinal tract as well as several lymph nodes also scored positive for capripox virus genome with low to moderate Cq-values (Table 3) [18], clearly indicating a generalized infection with GTPV after i.v. + s.c. or i.n. inoculation or i-c exposition. Considering clinical course and molecular data, GTPV-“V/103” represents a useful candidate for GTPV challenge model in future vaccine studies.

Two of three goats inoculated i.v. + s.c. or i.n. scored positive in the SNT, whereas both other goats of these groups remained negative in this assay until the day of euthanasia. Additionally, i-c goat Z/259 showed positive SNT result at 23 dpi. Contrarily, SNT of serum samples of i-c goat Z/260 was negative at all time points (Table 5).

The DA ELISA turned out to be less sensitive than the SNT but is also able to detect GTPV reactive antibodies. Positive results in the DA ELISA could only be detected in three out of eight animals, and in two out of three cases at a later time point than the SNT. Conclusively, examination of seroconversion is possible using both methods. Similar results were observed in cattle experimentally infected with an attenuated vaccine strain of LSDV and a LSDV field strain, respectively. Here, no difference in sensitivity and specificity could be seen comparing the SNT and the DA ELISA [39]. Nevertheless, the SNT turned out to be more sensitive for antibodies against GTPV than the DA ELISA in the present study. Therefore, the SNT is the gold standard for the analyses of a serological response of infected or vaccinated goats [7]. Despite its issues regarding the sensitivity, the DA ELISA provides a rapid and robust tool for serological examination on a herd level, having the advantage of avoiding work with infectious virus and allowing in addition high throughput testing.

### 4.3. Comparison of the Different Inoculation Methods

Three out of eight animals were either inoculated i.v. + s.c. or i.n. The remaining two animals were left as in-contact controls. Since no sheep showed clinical signs characteristic for sheep pox and viral genome could only be detected in two nasal swabs and a single oral swab sample shortly after intranasal inoculation, no conclusions can be drawn regarding the influence of the inoculation method of our SPPV animal trial.

However, in the GTPV group, variances in the incubation time for the different inoculation routes could be observed (Figure 2). Intravenously + subcutaneously inoculated goats had the shortest incubation time, showing clinical signs as early as 2 dpi. In contrast, incubation time for the intranasally inoculated goats took 8–10 days. In addition, the in-contact animals became also infected with GTPV via direct contact or aerosol. Here, first clinical signs were seen at day 20 pi and 24 pi. Differences in the severity of the clinical course (Figure 2), distribution of capripox virus genome in different organs (Table 3) or strength of serological response (Table 5) depending on the inoculation route were not detected. However, early and strong viremia could be associated with the i.v. + s.c. inoculation method, whereas viral shedding of i.n. inoculated goats seemed to be more prominent compared to the goats of the i.v. + s.c. group (Table 2). Additionally, the already described correlation between severity of clinical signs and length of viremia [18] was confirmed in our study.

## 5. Conclusions

In this study, we examined clinical signs, viral genome loads in four different matrices and serological responses after experimental infection of sheep and goats with SPPV-“V/104” strain and GTPV-“V/103” strain, respectively. While no clinical signs of sheep pox, no viremia, little viral shedding, and only slight serological response in one of eight sheep was detected, inoculation with GTPV-“V/103” led to a severe clinical course, high amounts of capripox virus genome in especially nasal and oral swabs as well as seroconversion in 5 out of 8 goats. The SPPV-“V/104” strain is likely a substantially attenuated SPPV vaccine strain. In contrast, the GTPV-“V/103” strain turned out to be highly virulent and pathogenic. Full genome sequencing using a combined Illumina short read and MinION long read approach confirmed this observation on the genetic level. Considering the molecular and serological data, nasal swabs provide a suitable, less-invasive, and sensitive tool for the detection of capripox virus genomes in infected animals. Additionally, the SNT proved to be the more sensitive and specific gold standard method for serological diagnostics.

## Figures and Tables

**Figure 1 viruses-12-01098-f001:**
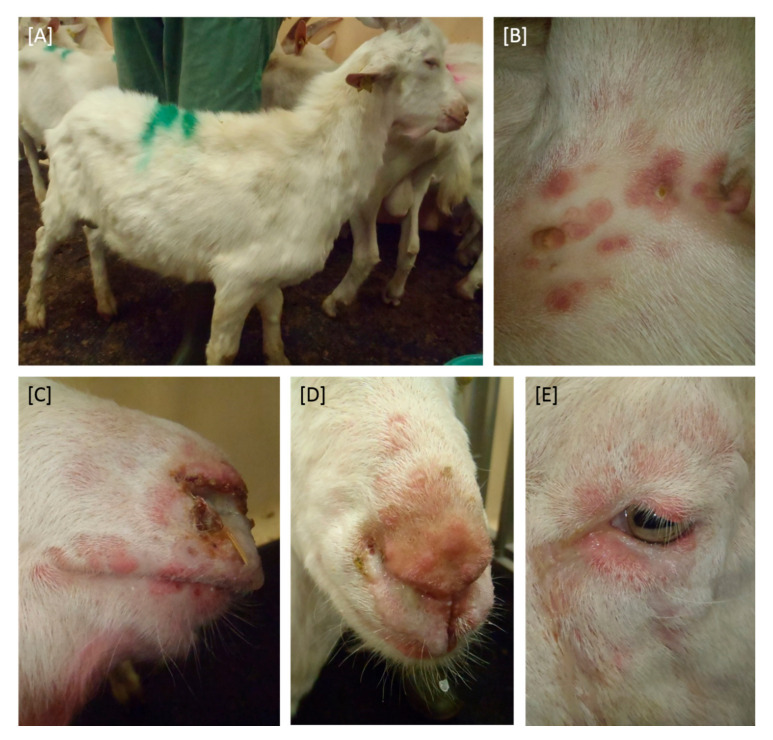
Clinical signs after inoculation with GTPV-“V/103“. All goats, independent whether inoculated intravenously and subcutaneously or intranasally as well as in-contact animals developed clinical symptoms characteristic for goat pox like lesions of the skin (**A**–**E**), nasal (**C**,**D**), and ocular discharge (**E**).

**Figure 2 viruses-12-01098-f002:**
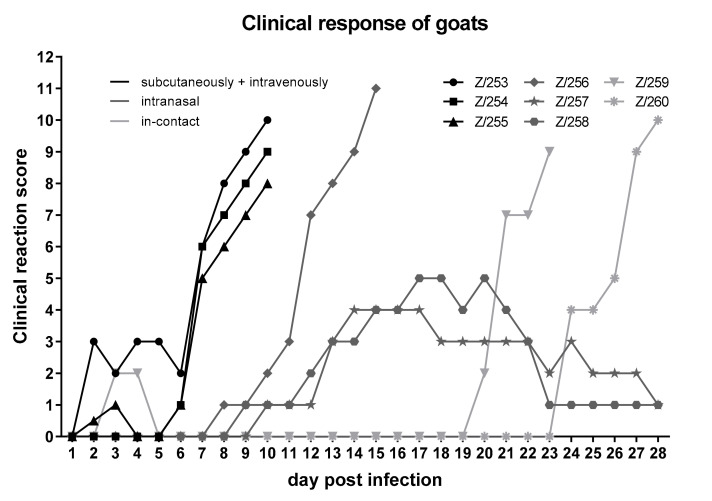
Clinical score of the goats inoculated with GTPV-“V/103”. Three goats were inoculated both intravenously and subcutaneously or intranasal with GTPV-“V/103“. Additionally, two in-contact goats were housed together with the inoculated animals.

**Figure 3 viruses-12-01098-f003:**
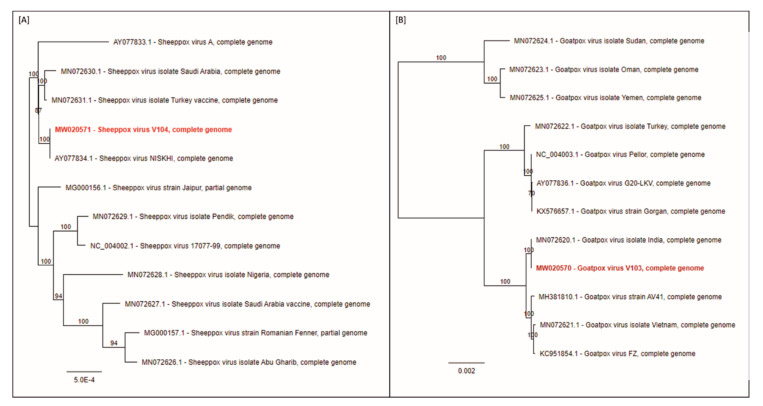
Phylogenetic trees displaying the relationship of the sequenced SPPV-“V/104“ and GTPV-“V/103“ strains. Nanopore and Illumina data were used in a hybrid assembly in an iterative mapping approach to gain full length, high quality consensus sequences. Phylogenetic relationship of (**A**) SPPV-“V/104“ and (**B**) GTPV-“V/103“ are presented.

**Table 1 viruses-12-01098-t001:** Results (Cq values) of EDTA-blood, serum, nasal, and oral swabs taken from sheep inoculated with SPPV-“V/104“ analyzed via the pan Capripox real-time qPCR.

SPPV-“V/104“		Capri-p32-Mix 1-Taq-FAM										
		dpi										
		0	3	5	7	10	13	15	18	21	23/24	28
Animal	matrix											
S/542 i.v. + s.c.	EDTA-blood	no Cq	no Cq	no Cq	no Cq	no Cq	no Cq	no Cq	no Cq	no Cq	no Cq	no Cq
	serum	no Cq	no Cq	no Cq	no Cq	no Cq	no Cq	no Cq	no Cq	no Cq	no Cq	no Cq
	nasal swab	no Cq	no Cq	no Cq	no Cq	no Cq	no Cq	no Cq	no Cq	no Cq	no Cq	no Cq
	oral swab	no Cq	no Cq	no Cq	no Cq	n.a.	no Cq	no Cq	no Cq	no Cq	no Cq	no Cq
S/573 i.v. + s.c.	EDTA-blood	no Cq	no Cq	no Cq	no Cq	no Cq	no Cq	no Cq	no Cq	no Cq	no Cq	no Cq
	serum	no Cq	no Cq	no Cq	no Cq	no Cq	no Cq	no Cq	no Cq	no Cq	no Cq	no Cq
	nasal swab	no Cq	no Cq	no Cq	no Cq	no Cq	no Cq	no Cq	no Cq	no Cq	no Cq	no Cq
	oral swab	no Cq	no Cq	no Cq	no Cq	n.a.	no Cq	no Cq	no Cq	no Cq	no Cq	no Cq
S/643 i.v. + s.c.	EDTA-blood	no Cq	no Cq	no Cq	no Cq	no Cq	no Cq	no Cq	no Cq	no Cq	no Cq	no Cq
	serum	no Cq	no Cq	no Cq	no Cq	no Cq	no Cq	no Cq	no Cq	no Cq	no Cq	no Cq
	nasal swab	no Cq	no Cq	no Cq	no Cq	no Cq	no Cq	no Cq	no Cq	no Cq	no Cq	no Cq
	oral swab	no Cq	no Cq	no Cq	no Cq	no Cq	no Cq	no Cq	no Cq	no Cq	no Cq	no Cq
S/652 i.n.	EDTA-blood	no Cq	no Cq	no Cq	no Cq	no Cq	no Cq	no Cq	no Cq	no Cq	no Cq	no Cq
	serum	no Cq	no Cq	no Cq	no Cq	no Cq	no Cq	no Cq	no Cq	no Cq	no Cq	no Cq
	nasal swab	no Cq	**36.4**	no Cq	no Cq	n.a.	no Cq	no Cq	no Cq	no Cq	no Cq	no Cq
	oral swab	no Cq	no Cq	**36.3**	no Cq	no Cq	no Cq	no Cq	no Cq	no Cq	no Cq	no Cq
S/673 i.n.	EDTA-blood	no Cq	no Cq	no Cq	no Cq	no Cq	no Cq	no Cq	no Cq	no Cq	no Cq	no Cq
	serum	no Cq	no Cq	no Cq	no Cq	no Cq	no Cq	no Cq	no Cq	no Cq	no Cq	no Cq
	nasal swab	no Cq	**38.0**	no Cq	no Cq	no Cq	no Cq	no Cq	no Cq	no Cq	no Cq	no Cq
	oral swab	no Cq	no Cq	no Cq	no Cq	n.a.	no Cq	no Cq	no Cq	no Cq	no Cq	no Cq
S/718 i.n.	EDTA-blood	no Cq	no Cq	no Cq	no Cq	no Cq	no Cq	no Cq	no Cq	no Cq	no Cq	no Cq
	serum	no Cq	no Cq	no Cq	no Cq	no Cq	no Cq	no Cq	no Cq	no Cq	no Cq	no Cq
	nasal swab	no Cq	no Cq	no Cq	no Cq	no Cq	no Cq	no Cq	no Cq	no Cq	no Cq	no Cq
	oral swab	no Cq	no Cq	no Cq	no Cq	no Cq	no Cq	no Cq	no Cq	no Cq	no Cq	no Cq
S/724 i-c	EDTA-blood	no Cq	no Cq	no Cq	no Cq	no Cq	no Cq	no Cq	no Cq	no Cq	no Cq	no Cq
	serum	no Cq	no Cq	no Cq	no Cq	no Cq	no Cq	no Cq	no Cq	no Cq	no Cq	no Cq
	nasal swab	no Cq	no Cq	no Cq	no Cq	no Cq	no Cq	no Cq	no Cq	no Cq	no Cq	no Cq
	oral swab	no Cq	no Cq	no Cq	no Cq	n.a.	no Cq	no Cq	no Cq	no Cq	no Cq	no Cq
S/725 i-c	EDTA-blood	no Cq	no Cq	no Cq	no Cq	no Cq	no Cq	no Cq	no Cq	no Cq	no Cq	no Cq
	serum	no Cq	no Cq	no Cq	no Cq	no Cq	no Cq	no Cq	no Cq	no Cq	no Cq	no Cq
	nasal swab	no Cq	no Cq	no Cq	no Cq	no Cq	no Cq	no Cq	no Cq	no Cq	no Cq	no Cq
	oral swab	no Cq	no Cq	no Cq	no Cq	n.a.	no Cq	no Cq	no Cq	no Cq	no Cq	no Cq

n.a.—displays sample not analyzable.

**Table 2 viruses-12-01098-t002:** Results (Cq values) of EDTA-blood, serum, nasal and oral swabs taken from goats inoculated with GTPV-“V/103“ analyzed via pan Capripox real-time qPCR.

GTPV-“V/103“		Capri-p32-Mix 1-Taq-FAM										
		dpi										
		0	3	5	7	10	13	15	18	21	23/24	28
Animal	matrix											
Z/253 i.v. + s.c.	EDTA-blood	no Cq	no Cq	36.4	34.3	32.1	-	-	-	-	-	-
	serum	no Cq	no Cq	no Cq	no Cq	37.5	-	-	-	-	-	-
	nasal swab	no Cq	no Cq	39.5	no Cq	27.9	-	-	-	-	-	-
	oral swab	no Cq	no Cq	no Cq	no Cq	34.8	-	-	-	-	-	-
Z/254 i.v. + s.c.	EDTA-blood	no Cq	no Cq	36.6	26.8	27.2	-	-	-	-	-	-
	serum	no Cq	no Cq	no Cq	37.7	34.9	-	-	-	-	-	-
	nasal swab	no Cq	no Cq	no Cq	37.0	26.6	-	-	-	-	-	-
	oral swab	no Cq	no Cq	no Cq	no Cq	32.2	-	-	-	-	-	-
Z/255 i.v. + s.c.	EDTA-blood	no Cq	no Cq	34.4	30.1	27.5	-	-	-	-	-	-
	serum	no Cq	no Cq	no Cq	no Cq	36.3	-	-	-	-	-	-
	nasal swab	no Cq	no Cq	36.1	no Cq	28.4	-	-	-	-	-	-
	oral swab	no Cq	no Cq	no Cq	no Cq	34.4	-	-	-	-	-	-
Z/256 i.n.	EDTA-blood	no Cq	no Cq	no Cq	no Cq	30.3	29.2	31.2	-	-	-	-
	serum	no Cq	no Cq	no Cq	no Cq	no Cq	35.6	37.9	-	-	-	-
	nasal swab	no Cq	no Cq	27.02	23.6	18.9	19.0	19.6	-	-	-	-
	oral swab	no Cq	no Cq	no Cq	28.6	27.4	26.0	25.7	-	-	-	-
Z/257 i.n.	EDTA-blood	no Cq	no Cq	no Cq	34.4	33.6	no Cq	no Cq	no Cq	no Cq	no Cq	no Cq
	serum	no Cq	no Cq	no Cq	n.a.	no Cq	no Cq	n.a.	no Cq	no Cq	no Cq	no Cq
	nasal swab	no Cq	37.7	26.3	21.4	19.9	19.1	20.4	21.5	22.4	26.6	26.1
	oral swab	no Cq	no Cq	36.9	32.0	27.8	27.7	26.4	30.2	29.6	32.1	36.1
Z/258 i.n.	EDTA-blood	no Cq	no Cq	no Cq	no Cq	33.0	35.0	no Cq	no Cq	no Cq	no Cq	no Cq
	serum	no Cq	no Cq	no Cq	no Cq	37.4	37.2	no Cq	no Cq	no Cq	no Cq	no Cq
	nasal swab	no Cq	38.1	29.4	24.0	24.0	19.8	21.5	25.4	25.0	24.4	33.4
	oral swab	no Cq	no Cq	36.8	33.6	31.2	31.0	28.9	29.8	32.7	31.2	35.7
Z/259 i-c	EDTA-blood	no Cq	no Cq	no Cq	no Cq	no Cq	no Cq	no Cq	no Cq	29.8	27.8	-
	serum	no Cq	no Cq	no Cq	no Cq	no Cq	no Cq	no Cq	no Cq	37.3	31.4	-
	nasal swab	no Cq	no Cq	no Cq	no Cq	no Cq	34.1	33.8	35.1	21.1	20.5	-
	oral swab	no Cq	no Cq	no Cq	no Cq	no Cq	34.4	36.0	33.4	33.6	30.3	-
Z/260 i-c	EDTA-blood	no Cq	no Cq	no Cq	no Cq	no Cq	no Cq	no Cq	no Cq	no Cq	35.0	29.6
	serum	no Cq	no Cq	no Cq	no Cq	no Cq	no Cq	no Cq	no Cq	no Cq	no Cq	35.9
	nasal swab	no Cq	no Cq	no Cq	no Cq	31.4	36.4	33.7	33.5	31.8	34.1	20.4
	oral swab	no Cq	no Cq	35.8	no Cq	37.3	33.3	37.8	31.7	33.9	33.9	27.9

-means animal was removed from the trial before sampling; n.a.—displays sample not analyzable.

**Table 3 viruses-12-01098-t003:** Results (Cq values) of particular organ samples taken during section from goats infected with GTPV-“V/103” analyzed using pan Capripox real-time qPCR.

GTPV-“V/103“	Capri p32 Taq Mix FAM							
Organ Sample	Intravenously + Subcutaneously			Intranasal			In-Contact	
	Z/253	Z/254	Z/255	Z/256	Z/257	Z/258	Z/259	Z/260
mandibular lymph node	30.4	33.4	34.1	31.0	35.1	no Cq	21.4	28.3
cervical lymph node	29.7	30.1	27.9	27.2	no Cq	no Cq	20.2	25.5
mediastinal lymph node	31.7	31.8	34.2	37.0	36.7	no Cq	23.1	32.2
mesenterial lymph node	36.5	no Cq	no Cq	no Cq	no Cq	no Cq	27.4	34.2
tonsil	36.2	no Cq	no Cq	32.0	no Cq	no Cq	24.8	28.3
spleen	32.9	36.2	35.3	31.5	no Cq	no Cq	19.8	32.0
liver	36.8	33.9	no Cq	35.5	no Cq	no Cq	22.8	33.7
lung	21.9	26.8	21.6	28.3	no Cq	no Cq	19.2	26.2
heart	no Cq	no Cq	no Cq	34.9	no Cq	no Cq	30.7	35.6
kidney	35.2	37.0	35.6	36.2	no Cq	no Cq	23.6	30.4
salivary gland	no Cq	no Cq	no Cq	no Cq	no Cq	no Cq	34.5	31.4
testicles	29.4	30.2	27.3	25.3	37.5	37.5	29.2	26.1
oral mucosa	36.9	32.3	32.1	27.0	31.3	33.4	23.5	28.4
thigh muscle	no Cq	37.0	no Cq	38.1	no Cq	no Cq	34.2	32.4
nose/conchae	n. t.	n. t.	n. t.	17.8	27.7	37.5	22.4	21.3
trachea	19.4	n. t.	26.2	21.6	28.1	n. t.	19.9	n. t.
rumen	n. t.	n. t.	33.0	n. t.	n. t.	n. t.	27.5	n. t.
abomasum	n. t.	n. t.	n. t.	n. t.	n. t.	n. t.	22.4	n. t.
epididymis	n. t.	n. t.	n. t.	18.5	n. t.	n. t.	n. t.	n. t.
location of skin sample								
mouth/nose	22.6	20.5	23.3	20.7	no Cq	36.6	20.7	21.4
eye/lid edge	25.3	22.3	25.0	20.2	no Cq	36.8	19.8	19.0
closely behind the ear	20.3	20.8	29.9	20.7	33.6	31.4	20.5	21.1
thoracic wall	18.9	20.8	29.6	21.5	33.0	21.3	21.2	19.1
skin inguinal	n. t.	23.5	25.1	18.8	n. t.	n. t.	n. t.	n. t.
hind leg	n. t.	n. t.	n. t.	n. t.	n. t.	21.3	n. t.	n. t.
tail base	n. t.	n. t.	n. t.	16.9	n. t.	n. t.	n. t.	n. t.
prepuce	n. t.	n. t.	n. t.	19.1	n. t.	n. t.	n. t.	n. t.
scrotum	22.3	n. t.	n. t.	n. t.	32.5	n. t.	24.4	19.8

n. t.—displays sample not taken.

**Table 4 viruses-12-01098-t004:** Serological analyses of serum samples taken from sheep inoculated with SPPV-“V/104“. For the ELISA, an S/P% of ≥30 was defined as positive. In the serum neutralization test (SNT), a titer ≥ 1:20 is defined positive.

SPPV-“V/104“		Serological Examination	
		ELISA (S/P%)	SNT (Titer)
S/542 i.v. + s.c.	0 dpi	−1	< 1:10
	7 dpi	14	1:13
	15 dpi	23	1:20
	21 dpi	51	1:25
	28 dpi	129	1:50
S/573 i.v. + s.c.	0 dpi	0	< 1:10
	7 dpi	3	< 1:10
	15 dpi	0	< 1:10
	21 dpi	0	1:13
	28 dpi	7	1:13
S/643 i.v. + s.c.	0 dpi	−1	< 1:10
	7 dpi	9	< 1:10
	15 dpi	1	< 1:10
	21 dpi	3	< 1:10
	28 dpi	3	< 1:10
S/652 i.n.	0 dpi	−1	< 1:10
	7 dpi	1	< 1:10
	15 dpi	4	< 1:10
	21 dpi	0	< 1:10
	28 dpi	0	< 1:10
S/673 i.n.	0 dpi	−1	< 1:10
	7 dpi	−1	< 1:10
	15 dpi	0	< 1:10
	21 dpi	0	< 1:10
	28 dpi	1	< 1:10
S/718 i.n.	0 dpi	−1	< 1:10
	7 dpi	−1	< 1:10
	15 dpi	−1	< 1:10
	21 dpi	−1	< 1:10
	28 dpi	−1	< 1:10
S/724 i-c	0 dpi	−1	< 1:10
	7 dpi	−1	< 1:10
	15 dpi	−1	< 1:10
	21 dpi	−1	< 1:10
	28 dpi	0	< 1:10
S/725 i-c	0 dpi	6	< 1:10
	7 dpi	−1	< 1:10
	15 dpi	−1	< 1:10
	21 dpi	0	< 1:10
	28 dpi	0	< 1:10

**Table 5 viruses-12-01098-t005:** Serological analyses of serum samples taken from goats inoculated with GTPV-“V/103“. For the ELISA, an S/P% of ≥30 was defined as positive. In the SNT, a titer ≥ 1:20 is defined positive.

GTPV-“V/103“		Serological Examination	
		ELISA (S/P%)	SNT (Titer)
Z/253 i.v. + s.c.	0 dpi	−1	<1:10
	7 dpi	−1	<1:10
	10 dpi	−1	1:13
Z/254 i.v. + s.c.	0 dpi	−1	<1:10
	7 dpi	0	1:13
	10 dpi	16	1:80
Z/255 i.v. + s.c.	0 dpi	0	<1:10
	7 dpi	0	<1:10
	10 dpi	53	1:25
Z/256 i.n.	0 dpi	−1	<1:10
	7 dpi	−1	<1:10
	15 dpi	23	1:16
Z/257 i.n.	0 dpi	−1	<1:10
	7 dpi	−1	<1:10
	15 dpi	11	1:50
	21 dpi	80	1:160
	28 dpi	83	1:80
Z/258 i.n.	0 dpi	−1	<1:10
	7 dpi	−1	<1:10
	15 dpi	1	1:80
	21 dpi	15	1:256
	28 dpi	65	1:128
Z/259 i-c	0 dpi	6	<1:10
	7 dpi	1	<1:10
	15 dpi	−1	<1:10
	21 dpi	−1	<1:10
	23 dpi	0	1:40
Z/260 i-c	0 dpi	−1	<1:10
	7 dpi	−1	<1:10
	15 dpi	−1	<1:10
	21 dpi	0	<1:10
	28 dpi	15	1:13

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
