# Peer review of "Experimental Infection and Genetic Characterization of Two Different Capripox Virus Isolates in Small Ruminants"

_viruses, 2020, doi:10.3390/v12101098_

Round 1

Reviewer 1 Report

In this paper examination of clinical signs, viral genome loads and serological response after experimental infection of sheep and goats were performed. Different inoculation routes were analyzed. Additionally full genome sequencing of two strains were made. The objective of the results, especially experimental infection,and the results are not really new. Additionally, despite the fact that in the title is "genetic characterization….” except phylogenetic trees no molecular analysis has been performed (for example sequence analysis).

Specific comments:

  1. Are the values given in the publication (viral load) related to CT values or to copy numer od viral DNA? The standard curve has been made?
  2. No description what method was used to make a phylogenetic tree.
  3. Lack accesion numbers of SPPV-„V/104” and GTPV-„V/103” strains.
  4. Figure „Clinical response of goats” is figure 2 not 1.
  5. No explanation of how the ”Clinical reaction score” (on Fig. 2) scale was evaluated.
  6. Line 209-210 and line 217. Why do you claim that Cq values between 36 and 38 is very high? Reactions are generally run for 40 cycles. Explain this.
  7. Improve the discussion. In most of the discussion, the author repeats the results described in the "results" section.

Author Response

Reviewer 1

In this paper examination of clinical signs, viral genome loads and serological response after experimental infection of sheep and goats were performed. Different inoculation routes were analyzed. Additionally full genome sequencing of two strains were made. The objective of the results, especially experimental infection, and the results are not really new. Additionally, despite the fact that in the title is "genetic characterization….” except phylogenetic trees no molecular analysis has been performed (for example sequence analysis).

Specific comments:

  1. Are the values given in the publication (viral load) related to CT values or to copy numer od viral DNA? The standard curve has been made?

Answer: For analyses of Capripox virus shedding and viremia, the semi-quantitative Cq values are mostly used, which is why we prepared Cq values in the manuscript. However, standard curve was generated, and copy number of viral DNA is available and can be included into supplemental material, when requested.

  • No description what method was used to make a phylogenetic tree.

Answer: We added this information in the manuscript section Material & Methods.

  • Lack accesion numbers of SPPV-„V/104” and GTPV-„V/103” strains.

Answer: Accession numbers were already requested when the manuscript was submitted and are now available and included into the manuscript.

  • Figure „Clinical response of goats” is figure 2 not 1.

Answer: We thank the reviewer for this comment and changed in in the manuscript.

  • No explanation of how the ”Clinical reaction score” (on Fig. 2) scale was evaluated.

Answer: We added the respective information into the manuscript.

  • Line 209-210 and line 217. Why do you claim that Cq values between 36 and 38 is very high? Reactions are generally run for 40 cycles. Explain this.

Answer: In moderately and severely affected animals, Cq values of nasal and oral swab samples are generally lower (around 20-25), as this could be seen for the goats in this study. In addition, this could also be seen for cattle infected with LSDV and sheep affected by sheep pox virus (which is published by several working groups and which we could validate during a recent animal trial). Compared to these findings, Cq values of 35-40 are relatively high, representing a low viral load in these samples.

  • Improve the discussion. In most of the discussion, the author repeats the results described in the "results" section.

Answer: We thank the reviewer for this comment and we deleted extraneous results from the discussion.

Reviewer 2 Report

Wolff et al have examined the pathology and followed viremia as well as the immune response in sheep and goats experimentally infected either with a sheeppox strain (in sheep) or a goatpox strain (in goats). The study is clearly presented and the conclusions drawn are overall supported by the data. It turns out that the sheeppox strain is not pathogenic in sheep, inducing practically no viremia and hardly any apparent immune response whereas the goatpox strain is highly pathogenic leading to death of a number of animals and severe morbidity in the others. Infected goats display the characteristic symptoms of classical goatpox, clear-cut viremia and develop a low but significant level of antibodies according to the assays used. The authors have also determined the nucleotide sequences of the two viral genomes and compared them to related viral sequences in public databases. Sequence data suggests that the sheeppox isolate is phylogenetically close to a previously characterized sheeppox vaccine strain whereas the goatpox isolate is identical to a previously characterized virulent goatpox strain. One could argue that if the authors had begun their study by sequencing the viral genomes they would have rapidly concluded that the goatpox isolate in their hands is a virulent strain. Nevertheless, their study does provide some useful information for future vaccination trials they suggest they might attempt. On the other hand, the data presented and the discussion do not allow one to definitively conclude as to the phenotype of the sheeppox strain. As the authors have used cross-bred sheep one cannot exclude the possibility that one of the sheep varieties or even both display natural resistance to sheeppox that is not found in other sheep varieties. In any case the authors should state, if it is known, the parents of the cross-breed. A little bit more detail should also be provided regarding the sequencing data for the sheeppox strain. For instance, the authors mention that their sequence is 99.9 % identical to a well-known vaccine strain (NISKHI). Actually data in the literature indicate that the NISKHI strain is also 99.9 % identical to at least one other virulent strain (Tulman et al. 2002). So it would be interesting if the authors could provide some information about which genes in their sheeppox strain vary relative to both the virulent strain and the NISKHI vaccine strain. Do they mostly involve the same set of gene changes that occur in the previously sequenced vaccine strain or do they involve another set of genes or alternatively are the changes simply spread out over the genome without appearing to affect any particular set of genes? This analysis would strengthen (or weaken) their hypothesis that their sheeppox strain is a highly attenuated strain, whatever the sheep used as hosts. Another concern is the poor sensitivity of the ELISA test used for titrating capripoxvirus antibodies. It’s unclear exactly what viral antigens this test employs. The authors should provide as much information as possible on this topic. An ELISA developed in the authors’ own laboratory using purified virus or infected cells might have been much more sensitive. 

Minor comments:

Line 23 replace “was” by “were”

Paragraph 2.2 Please indicate the cell line used to grow the virus stocks for inoculations and how the virus suspensions for inoculation were prepared (crude or pure virus, excipient used).

Line 173 replace “all” by “any”

Line 406 replace “less” by “little”

Author Response

Reviewer 2

Wolff et al have examined the pathology and followed viremia as well as the immune response in sheep and goats experimentally infected either with a sheeppox strain (in sheep) or a goatpox strain (in goats). The study is clearly presented and the conclusions drawn are overall supported by the data.

Answer: We thank the reviewer for this kind assessment of our work.

It turns out that the sheeppox strain is not pathogenic in sheep, inducing practically no viremia and hardly any apparent immune response whereas the goatpox strain is highly pathogenic leading to death of a number of animals and severe morbidity in the others. Infected goats display the characteristic symptoms of classical goatpox, clear-cut viremia and develop a low but significant level of antibodies according to the assays used. The authors have also determined the nucleotide sequences of the two viral genomes and compared them to related viral sequences in public databases. Sequence data suggests that the sheeppox isolate is phylogenetically close to a previously characterized sheeppox vaccine strain whereas the goatpox isolate is identical to a previously characterized virulent goatpox strain. One could argue that if the authors had begun their study by sequencing the viral genomes they would have rapidly concluded that the goatpox isolate in their hands is a virulent strain. Nevertheless, their study does provide some useful information for future vaccination trials they suggest they might attempt.

Answer: This is correct, genetic data reveal that the used SPPV strain highly likely is a vaccine strain and the GTPV isolate is a virulent strain. Nevertheless, we wanted to analyze possible adverse effects after vaccination and wanted to study pathogenesis of the virulent GTPV strain for establishment of a challenge model, as the reviewer already wrote. An additional aim of both studies was generation of reference material, at least of positive sera of sheep and goats, for internal validation of diagnostic methods.

On the other hand, the data presented and the discussion do not allow one to definitively conclude as to the phenotype of the sheeppox strain. As the authors have used cross-bred sheep one cannot exclude the possibility that one of the sheep varieties or even both display natural resistance to sheeppox that is not found in other sheep varieties. In any case the authors should state, if it is known, the parents of the cross-breed.

Answer: Sheep used in this study were bought from local breeders in Germany. Unfortunately, parental breeds are unknown. We added a note into the manuscript about the origin of the parental breeds.

We fully agree with the reviewer that natural resistance is possible. However, different studies dealing with experimental infection of sheep, goats and cattle with Capripox viruses show that European breeds are more susceptible to Capripox virus infections than indigenous breeds. In addition, we recently performed another animal trial in which sheep of comparable breed were inoculated with two different virulent SPPV strains. In all groups, animals got severely affected by sheeppox, indicating susceptibility of German breeds towards SPPV. Since we used comparable sheep in the present study, we would expect reaction in at least few animals after experimental inoculation, even if some of the sheep were naturally resistant.

A little bit more detail should also be provided regarding the sequencing data for the sheeppox strain. For instance, the authors mention that their sequence is 99.9 % identical to a well-known vaccine strain (NISKHI). Actually data in the literature indicate that the NISKHI strain is also 99.9 % identical to at least one other virulent strain (Tulman et al. 2002). So it would be interesting if the authors could provide some information about which genes in their sheeppox strain vary relative to both the virulent strain and the NISKHI vaccine strain. Do they mostly involve the same set of gene changes that occur in the previously sequenced vaccine strain or do they involve another set of genes or alternatively are the changes simply spread out over the genome without appearing to affect any particular set of genes? This analysis would strengthen (or weaken) their hypothesis that their sheeppox strain is a highly attenuated strain, whatever the sheep used as hosts.

Answer: We fully agree with the reviewer that the sequence identities of SPPV attenuated vaccine and virulent field strains are very high. In contrast to LSDV the genetic markers, important for the attenuation of the SPPV vaccine strains, are not very consistent. In addition, not all sequenced SPPV strains can be clearly defined as vaccine or field strain based on the according published data. Recently, Biswas and co-workers (Biswas et al., 2020) analysed the currently available and novel full genome sequence data of capripox viruses also for the mutants potentially involved in virus strain attenuation. Here it could be shown, that the Ankyrin repeat protein genes 145 and 148 are truncated in the SPPV-NISKHI vaccine strain. Similar truncated genes 145 and 148 could be also identified in our sequenced SPPV-V104. This finding, together with the results of the in-vivo study, support the classification of this strain in the group of the attenuated SPPV vaccines.

We added the respective information into the manuscript.

Another concern is the poor sensitivity of the ELISA test used for titrating capripoxvirus antibodies. It’s unclear exactly what viral antigens this test employs. The authors should provide as much information as possible on this topic. An ELISA developed in the authors’ own laboratory using purified virus or infected cells might have been much more sensitive. 

Answer: This is correct, the used DA ELISA turned out to be less sensitive than the SNT for GTPV. Since this ELISA is commercially available, viral antigens are industrial secrets of ID.vet. In our study, the aim was to compare this commercially available method that provides some advantages (e.g. faster results, no usage of live virus) in comparison to the SNT with the actual gold standard for serological testing (SNT). An in-house ELISA would have given higher sensitivity, but would not provide advantages for detection of antibodies against SPPV in the field.

Minor comments:

Line 23 replace “was” by “were”

Line 173 replace “all” by “any”

Line 406 replace “less” by “little”

Answer: We thank the reviewer for the corrections of the English language and changed the respective words in the text.

Paragraph 2.2 Please indicate the cell line used to grow the virus stocks for inoculations and how the virus suspensions for inoculation were prepared (crude or pure virus, excipient used).

Answer: We added the respective information regarding propagation of virus and how the inoculation material was prepared into the text.